# Ageing-Related Alterations in Renal Epithelial Glucose Transport

**DOI:** 10.3390/ijms242216455

**Published:** 2023-11-17

**Authors:** Chien-Te Lee, Hwee-Yeong Ng, Hua-Rong Zhong, Yi Wang, Chih-Han Liu, Yuai-Ting Lee

**Affiliations:** 1Division of Nephrology, Department of Internal Medicine, Kaohsiung Municipal Feng-Shan Hospital (Under Management of Chang Gung Medical Foundation), Kaohsiung 83062, Taiwan; gvboy@cgmh.org.tw (H.-R.Z.); dlm0909@cgmh.org.tw (C.-H.L.); yuai@cgmh.org.tw (Y.-T.L.); 2Division of Nephrology, Department of Internal Medicine, Kaohsiung Chang Gung Memorial Hospital and Chang Gung University College of Medicine, Kaohsiung 83301, Taiwan; kujiben@gmail.com (H.-Y.N.); wangea@cgmh.org.tw (Y.W.)

**Keywords:** ageing, hydrogen sulfide, GYY4137, glucose transporter, glucose homeostasis

## Abstract

The kidney plays a crucial role in glucose homeostasis by regulating glucose transport. We aimed to investigate the impact of alterations in glucose transport on glucose metabolism during ageing. Adult male Sprague Dawley rats were divided into five groups: 3-month, 6-month, and 12-month control groups, and 6- and 12-month groups receiving the hydrogen sulfide donor molecule GYY4137. The study found that, as age increased, daily urinary uric acid and protein levels increased in the 12-month group. Blood sugar level and HOMA-IR index increased in the 12-month group, and were partially improved by GYY4137. The kidney tissue showed mild glomerulosclerosis in the 12-month group, which was diminished by GYY4137. Gene expression analysis showed decreased sirtuin and increased p21 expression in the aging groups. Increased SGLT1 and SGLT2 expression was observed in the 12-month group, which was reversed by GYY4137. Both GLUT1 and GLUT2 expression was increased in the 6- and 12-month groups, and reversed by GYY4137 in the 12-month group. The study concluded that aging was associated with increased blood sugar levels and the HOMA-IR index, and the abundance of renal glucose transporters increased as aging progressed. GYY4137 effectively reversed aging-related alterations in glucose homeostasis and renal epithelial transporters.

## 1. Introduction

Ageing is considered an irreversible, natural process that leads to death [1]. The so-called age-related diseases include the most notably chronic diseases such as heart disease, stroke, Alzheimer disease, diabetes, chronic obstructive pulmonary disease, chronic kidney disease, osteoporosis, arthritis and even cancers. Nevertheless, the mechanisms underlying their initiation and progression remain largely unknown. With the aging process, numerous metabolic pathways are disturbed and deranged, further causing diseases and dysfunction [2]. It remains a challenge to counteract this irreversible and progressive process. A variety of interventional strategies have been developed as anti-ageing therapy, including calorie restriction, exercise and even pharmacological treatments [3].

It is recognized that ageing is accompanied by increased blood glucose and insulin levels [4,5]. An epidemiological study indicated that the prevalence of type 2 diabetes increases with age. It has been estimated that more than 80% of the elderly have diabetes or impaired glucose metabolism, and the incidence of diabetes is growing rapidly for people over the age of 65 [6]. The underlying mechanisms responsible for the changes in glucose metabolism during ageing are multiple. Increased insulin resistance, decreased peripheral glucose uptake, impaired β-cell function, hypomagnesemia and changed body composition are major contributors to abnormal glucose metabolism [7]. Among them, impaired β-cell function and impaired β-cell adaptation to insulin resistance with resultant impaired insulin secretion are considered the critical factors. Both animal and clinical studies have found that the ageing process is associated with dysregulated β-cell turnover and function [8]. Ageing per se reduces the β-cell regeneration capacity. The insulin secretion in response to glucose is progressively decreased in the elderly. Furthermore, reduced physical activity, obesity and loss of skeletal muscle all are important factors that are responsible for insulin resistance in this population.

The kidney plays a distinct role in glucose homeostasis. In normal situations, four major metabolic processes occurring in the kidney contribute to glucose homeostasis. The kidney is an important extra-hepatic organ responsible for gluconeogenesis. It is estimated that renal gluconeogenesis accounts for 40% of total gluconeogenesis [9,10]. Approximately 180 gm glucose is freely filtered by glomeruli each day. To regain glucose, tubular reabsorption takes place and most of the filtered glucose traverses the epithelial cells into circulation. Three renal glucose transport molecule systems have been identified: SLC2, SLC5: sodium substrate symporter family (SSSF), and the encoded by SCL50 gene (SWEET) protein. Renal epithelial cells express one or more glucose transporter. These transporters exhibit specific regulatory properties and functions [11]. Urinary excretion is a physiological route for glucose excretion. In normal circumstances, a minimal amount of glucose is excreted in the urine. Genetic defects in these transporters, such as renal Fanconi syndrome, have been identified with phenotypic manifestations of increased glucosuria [12]. The novel application of the sodium–glucose co-transporter 2 (SGLT2) inhibitor has revolutionized our treatment of diabetes mellitus. By enhancing urinary glucose excretion, pharmacological intervention with SGLT2 inhibitors is an effective and safe therapy for glycemic control [13]. Nevertheless, the role of the kidney in ageing-related changes in glucose metabolism has rarely been investigated, and whether renal epithelial glucose transporters are affected remains unclear.

Hydrogen sulfide is a toxic gas characterized by its strong odor of rotten eggs. Previous studies on hydrogen sulfide have mainly focused on its toxicity. It was not until recently that physiological studies discovered the role that hydrogen sulfide plays in biological systems [14]. Hydrogen sulfide is recognized as an important signaling molecule with various physiological effects and has been shown to be involved in cardiovascular system diseases, cancer, and neurodegenerative diseases [15]. The most widely studied functions of endogenous hydrogen sulfide are its vasodilatory effects and its ability to reduce and modulate oxidative stress [16]. More recently, studies have indicated the therapeutic potential of hydrogen sulfide in renal diseases, including acute and chronic kidney disease models. The key metabolic enzymes of hydrogen sulfide in the kidney were reduced, which led to a decrease in hydrogen sulfide. The administration of hydrogen sulfide can alleviate and reverse disease development and progression. The effects of hydrogen sulfide in renal tissue are multiple. In addition to ROS, it can inhibit the renin–angiotension–aldosterone system. Different kinds of renal cells are affected, leading to significant alterations in structure and function [17]. Nevertheless, the direct effect of hydrogen sulfide on renal epithelial transport has rarely been studied.

In the present study, we aimed to investigate the ageing-related alterations in the renal epithelial glucose transporters SGLT and GLUT. GYY4137, a slow-releasing hydrogen sulfide donor, was introduced to reverse aging-associated changes.

## 2. Results

### 2.1. Laboratory Data

Animals were divided into five groups: the control group (3 months), 6 months, 6 months with GYY4137 treatment, 12 months, and 12 months with GYY4137 treatment. As shown in Table 1, their body weight increased with time. A significant increase was noted in animals of the 12-month control group and in the 12-month group with GYY4137 treatment compared to the 3-month group. There was no significant difference in the daily urine amount among all groups. Blood glucose levels were increased in the 6- and 12-month groups compared to the 3-month group animals. Blood concentrations of insulin significantly increased in the 12-month groups compared to the 3-month group and did not change in the 6-month groups (2.5 ± 1.0 vs. 3.2 ± 1.9 vs. 4.4 ± 1.1 μU/mL, *p* < 0.05). Treatment with GYY4137 did not affect insulin levels. The HOMA-IR index was similar between the 3- and 6-month groups and increased significantly in the 12-month group compared to the 3-month group and 6-month group, respectively (0.7 ± 0.2 vs. 1.2 ± 0.3; 0.6 vs. 1.2 ± 0.3; both *p* < 0.05). Treatment with GYY4137 reduced the HOMA-IR index in the 12-month groups (1.2 ± 0.3 vs. 0.9 ± 0.2, *p* < 0.05). There was a modest increase in serum creatinine levels in the 6- and 12-month age groups. Daily urine glucose excretion did not differ among the groups. A significant increase in daily uric acid excretion was noted in the 12-month groups (23.9 ± 6.7 vs. 16.2 ± 3.8 mg/day, *p* < 0.05). GYY4137 treatment did not affect uric acid excretion in both the 6- and 12-month groups. Daily urinary protein excretion was similar between the 3- and 6-month groups, and there was a significant increase in the 12-month groups. GYY4137 treatment was associated with reduction in proteinuria (52.5 ± 18.2 vs. 87.1 ± 24.4, *p* < 0.05).

### 2.2. Blood Glucose Levels in Pyruvate Tolerance Test

The pyruvate tolerance test represents hepatic gluconeogenesis. As shown in Figure 1, in 3-month group animals, blood glucose levels were 112.6 ± 7.4 mg/dL at the beginning, 193.8 ± 30.0 mg/dL at 30 min, 200.1 ± 25.8 mg/dL at 60 min, 174.5 ± 27.2 mg/dL at 90 min, and 155.4 ± 23.4 mg/dL at 120 min. They were 115.0 ± 22.0 mg/dL, 249.0 ± 70.0 mg/dL, 244.0 ± 60.2 mg/dL, 211.1 ± 65.7 mg/dL and 187.5 ± 47.2 mg/dL in the 6-month group. In the 6-month group with GYY4137 treatment, blood sugar levels were 91.8 ± 12.2 mg/dL, 173.6 ± 26.0 mg/dL, 176.4 ± 38.7 mg/dL, 152.1 ± 35.0 mg/dL, and 138.6 ± 26.4 mg/dL. In the 12-month group, they were 111.6 ± 18.8 mg/dL, 166.4 ± 34.8 mg/dL, 186.0 ± 66.4 mg/dL, 149.9 ± 29.6 mg/dL, and 142.6 ± 25.4 mg/dL. In the 12-month group with GYY4137 treatment, they were 105.8 ± 4.7 mg/dL, 211.6 ± 58.0 mg/dL, 212.0 ± 86.5 mg/dL, 188.0 ± 86.7 mg/dL, and 175.5 ± 70.9 mg/dL. There was no difference between the four time points (30, 60, 90 and 120 min) in the measurements of each group (*p* > 0.050). A comparison analysis with/without GYY4137 treatment revealed significant differences in 30 and 60 min measurements in the 6-month groups (*p* < 0.05). No difference was noted in the 12-month groups with/without GYY4137 treatment.

### 2.3. Histology Study of Renal Tissue (H & E Staining, N = 4 in Each Group, Figure 2)

As displayed in Figure 2, there were no significant changes in the glomeruli structure in the 6-month group compared with the 3-month group, and a modest increase in glomerulosclerosis was noted in the 12-month group. GYY4137 treatment in the 12-month group was associated with a modest decrease in glomerulosclerosis (Table 2).
Figure 2H & E staining of renal issue of all study-group animals. Severity of glomerulosclerosis graded from 0 to 4. (upper panel: 400×; lower panel: 100×).
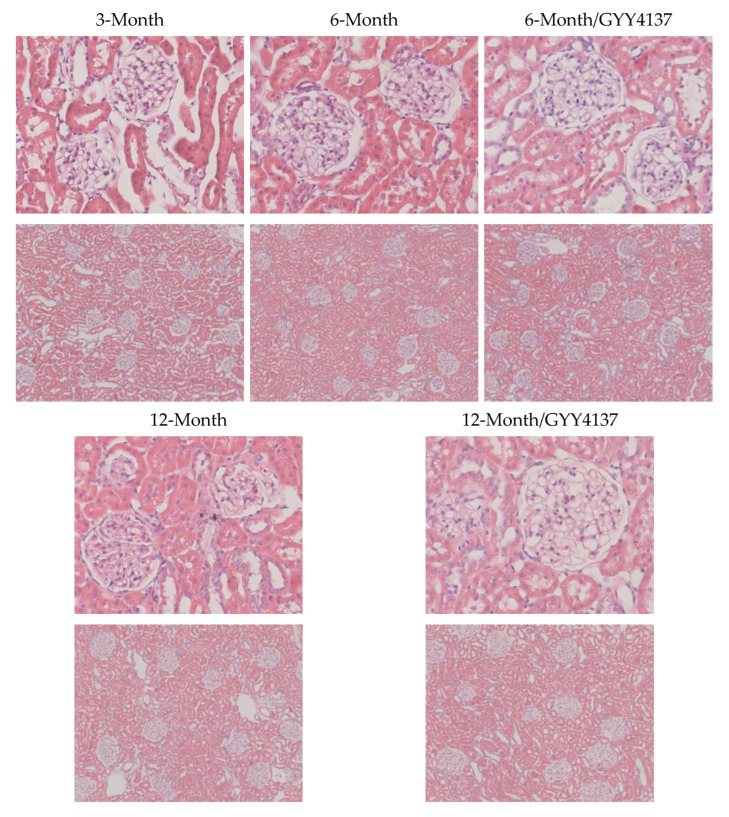



### 2.4. Gene Expression Analysis (N = 6 in Each Group, Figure 3)

The expression of SIRT1 was similar between the 3- and 6-month groups (88 ± 6.0% of control), and GYY4137 treatment did not affect the expression (101.7 ± 10.2% of control). The expression was decreased in the 12-month group (70.1 ± 6.8% vs. 3 months) and GYY4137 treatment reversed the decrease (86.3 ± 9.7% of control). The expression of P21 was significantly increased in the 6-month group (228.8 ± 12.1% of control) and GYY4137 treatment reduced the increase (155.1 ± 13.6% of control). The expression was also increased in the 12-month group (267.5 ± 10.8% of control) and treatment with GYY4137 reduced the increase (160.0 ± 9.6% of control).
Figure 3Gene expression of SIRT1 and P21 in 3-month, 6-month, 6-month with GYY4137 treatment (G6M), 12-month and 12-month with GYY4137 treatment (G12M) groups. * *p* < 0.05 vs. 3-month group, ^#^ *p* < 0.05: 6-month group vs. G6-month group. ^##^ *p* < 0.05: 12-month group vs. G12-month group.
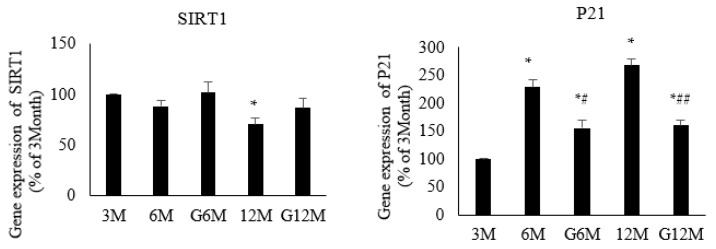


### 2.5. Immunoblotting (N = 6 in Each Group, Figure 4)

Figure 4 displays the results of immunoblotting in the present study. The abundance of klotho was similar across the five groups. There was no significant change in the 6-month group, 6-month group with GYY4137 treatment, 12-month group, or 12-month group with GYY4137 treatment (99.8 ± 3.9% of control). Compared with the 3-month group, SGLT1 was increased in the 6-month group (130.0 ± 1.4% of control) and GYY4137 treatment diminished the increase (112.6 ± 1.6% of control). The abundance of SGLT2 was unchanged in the 6-month group (115.1 ± 1.9% of control), but was significantly increase in the 12-month group (135.7 ± 2.1 of control, *p* < 0.05) and was reversed by GYY4137 (109.3 ± 3.1% of control). GLUT1 was increased in both the 6- and 12-month groups (113.3 ± 1.3% of control and 109.5 ± 1.7% of control; both *p* < 0.05), and this was reversed by GYY4137 treatment in the 12-month group (93.0 ± 1.9% of control). The abundance of GLUT2 was increased in both the 6- and 12-month groups (140.6 ± 0.8% of control and 138.9 ± 1.6% of control; both *p* < 0.05), and GYY4137 treatment reversed the increase in the 12-month group (102.7 ± 3.6% of control).
Figure 4Protein expression of klotho, SGLT1, SGLT2, GLUT1 and GLUT2 in renal tissue of control, 6-month, 6-month with GYY4137 treatment (G6M), 12-month and 12-month with GYY4137 treatment (G12M) groups. * *p* < 0.05 vs. 3-month group, ^##^ *p* < 0.05: 12-month group vs. G12-month group.
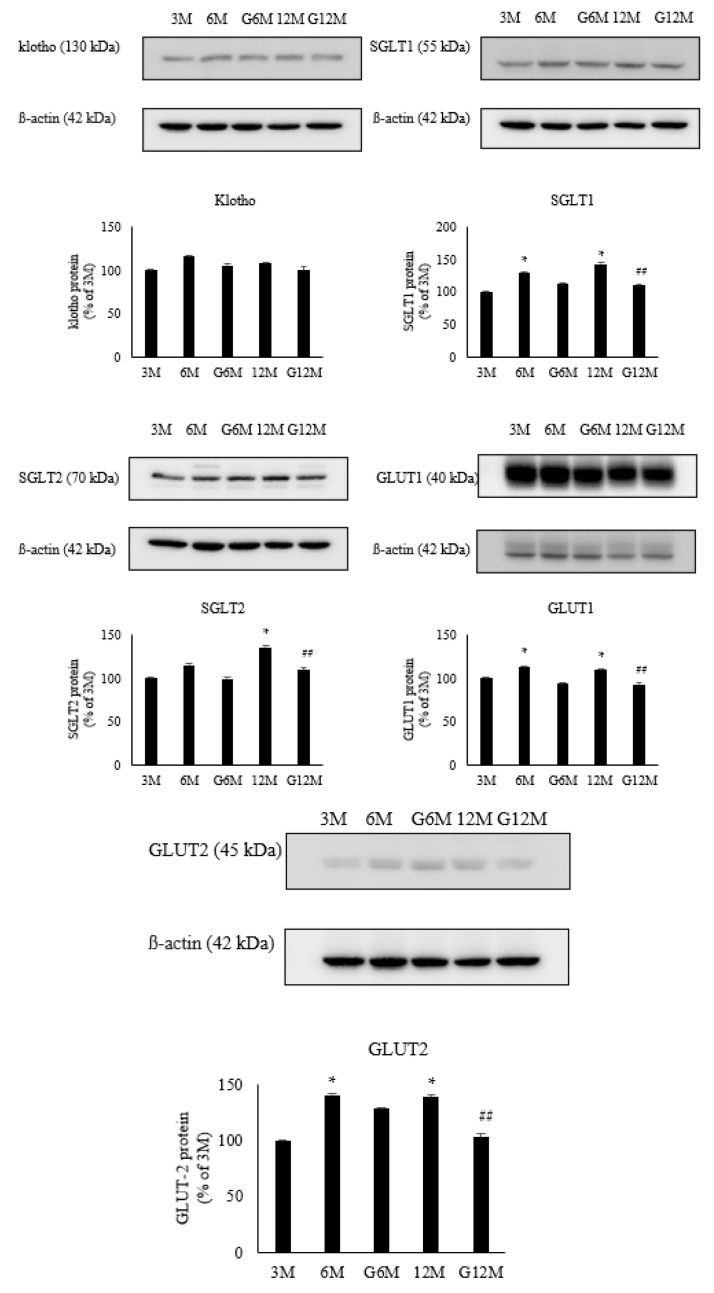


## 3. Discussion

Our results clearly indicated that ageing was associated with an increased expression of SGLT1, SGLT2, GLUT1, and GLUT2 in the kidney. With ageing, both blood sugar and insulin levels increased, which contributed to the increase in the HOMA index. The administration of GYY4137 reversed the increased expression of glucose transporters, in addition to decreasing the insulin resistance index.

Biochemical studies revealed that, as ageing progressed, the HOMA-IR index increased. The increase was remarkable in the 12-month group of animals but not in the 6-month group. Blood glucose levels were increased in 6-month and 12-month groups. This finding is comparable with other studies showing the contribution of increased glucose and insulin levels as ageing progresses. It has been reported that a 1–2 mg% increase in fasting and 15 mg% increase in postprandial blood glucose levels occurred in humans [18]. The results of the pyruvate tolerance test in the present study indicated that hepatic glucose production was not affected during the ageing process. This finding was similar to previous clinical studies [6] suggesting that gluconeogenesis in the liver is not affected as age increases. GYY4137 did not have impactful effects on hepatic gluconeogenesis in the ageing process. The administration of GYY4137 did not normalize but partially reversed the increase in blood glucose, insulin levels and HOMA-IR index in the 12-month group. We found there was no significant change in daily urinary glucose excretion in the 6- and 12-month groups compared with the 3-month group. As the renal threshold for glucose increases with age, glucosuria is not evident in the elderly, and thus polyuria and polydipsia may not become obvious in the elderly [18]. This finding suggests that clinicians should pay attention to symptoms of diabetes in the aged population.

The role of hydrogen sulfate in glucose metabolism has been investigated in recent decades [19,20,21]. In the pancreas, hydrogen sulfide can either inhibit or enhance insulin synthesis/section in β-cells. The reason for these contradictory results is not clear [20]. In the liver, a hermetic theory was proposed, as the low levels of hydrogen sulfide that promote gluconeogenesis are beneficial but high levels that suppress gluconeogenesis may be harmful [22]. Hydrogen sulfide also affects glucose uptake in the adipocyte. Moreover, it has been shown that hydrogen sulfide reduced the production of inflammatory cytokines in adipose tissue [23]. Skeletal muscles represent a major organ for glucose utilization and storage in glucose metabolism. Several in vivo studies have shown that hydrogen sulfide can improve glucose metabolism in skeletal muscle and reduce diabetic-associated atrophy and sarcopenia [24]. In addition to insulin section, Pichette et al. demonstrated that hydrogen donors NaHS and GYY4137 can both directly stimulate GLP-1 section in vitro, leading to an up to two-fold increase in a dose-dependent manner [25]. They explored the molecular mechanism by which hydrogen sulfide exerts this biological effect via the activation of p38 mitogen-activated protein kinase. Taken together, hydrogen sulfide affects important metabolic organs such as the pancreas, liver, skeletal muscle, and adipose tissue, and thus modulates glucose homeostasis. In one recent community study, serum sulfide level and glycemic status were analyzed. A low sulfide level was associated with impaired glucose intolerance [26]. Moreover, accumulating evidence has proven the therapeutic potential of hydrogen sulfide in preventing or treating diabetic complications such as cardiomyopathy, retinopathy, nephropathy, vasculopathy and encephalopathy through its anti-apoptosis and anti-fibrosis effects, as well as its inhibition of oxidative stress and inflammation [21,27].

Renal epithelial glucose transport plays an important role in maintaining glucose homeostasis. In the present study, we found that the abundance of SGLT1 and SGLT2 was increased in the 12-month group animals. Both SGLT1 and SGLT2 are members of the SLC5 family. SGLT2 is a major glucose transporter, contributing to the 90% reabsorption rate of filtered glucose. Previous studies have clearly demonstrated the upregulation of SGLT1 and SGLT2 in metabolic syndrome and diabetes [28]. Recent studies have indicated that, through the pharmacological inhibition of SGLT2, diabetes can be controlled and additional cardiovascular and renal benefits can be achieved [29]. We found that the administration of GYY4137 was effective in reducing the increase in SGLT1 and SGLT2 during the aging process. These changes were associated with a decrease in the blood concentration of glucose and insulin, and HOMA-IR. Changes in the glucose transporters GLUT1 and GLUT2 were also examined. These two transporters are expressed on the basolateral site of renal epithelial cells and could act as facilitators of glucose entry into the blood circulation. Similar to SGLT1/2, our results indicated a significant increase in GLUT1/2 during the aging process. This increase was diminished by GYY4137 in 12-month group animals. Very few studies have investigated the effects of hydrogen sulfide on renal epithelial transportation. It has been reported in physiological experiments that exogenous hydrogen sulfide inhibited the sodium–potassium–chloride co-transporter and enhanced the urinary excretion of sodium and potassium [17]. By regulating renal epithelial transport machinery, such as SGLT2, the kidney exhibits the ability to maintain homeostasis in its acid base status, electrolytes, and fluid control, and, in this case, glucose metabolism. These findings illustrate the distinct physiological and pathophysiological role of the kidney [30,31,32].

The renal manifestations of the ageing process include several other aspects, in addition to structural changes. A decrease in nephron number and size, glomerulosclerosis, and extracellular matrix accumulation are characteristics of renal ageing [33]. All components of renal tissue present ageing-related changes, such as glomeruli, tubulointerstitium, and vasculature. Overall, ageing leads to a decline in renal function. In the present study, although the amount of creatinine in blood levels increased, creatinine clearance was not decreased in the 6- and 12-month groups. A histological examination also revealed that age-associated glomerulosclerosis was not remarkable in the 12-month group. Therefore, the effects of GYY4137 were not apparent in renal tissue structures, at least in glomeruli. Other tubular dysfunctions include decreased sodium reabsorption, altered transcellular potassium gradient, impaired urinary concentration, and response to fludrocortisone [34]. All of these changes result in clinically susceptible circumstances that warrant further attention [35]. The interaction between all components of these changes is largely unclear. Renal cellular senescence plays an important role in renal ageing. It has been reported that tubular cells were the most susceptible to senescence among all renal cell types [36]. Several aging pathways are responsible for this biological process in the kidney, including increased oxidative stress, increased angiotensin II sensitivity, inflammation, and fibrosis. The two main signaling pathways implicated in the ageing process are p53/p21 and p16/Rb [37]. P21 has long been considered a marker of senescence. Our study indicated an increased gene expression of p21 in renal tissue as ageing progressed, but the administration of GYY4137 reduced the increase. This finding illustrates that ageing-associated senescence can be partially reversed by GYY4137. The klotho protein was identified decades ago, and, in combination with FGF23 on the FGF receptor, forming the so-called FGF–Klotho endocrine axis, plays a critical role in calcium/vitamin D/parathyroid hormone homeostasis. Klotho is expressed in the distal convoluted tubule of the kidney. The premature ageing characteristics of klotho-knockout animals illustrate the critical role of klotho in the ageing process [38]. It has been reported that klotho can reduce senescence in epithelial cells [39] and suppress the insulin and IGF1 signaling pathways in animal studies [40,41]. However, the direct effects underlying klotho-associated aging remain unclear. In the present study, we did not find a significant change in klotho expression in the renal tissue of the 6- and 12-month group animals. It is speculated that the study animals were not as old as those used to demonstrate alterations in renal klotho expression.

Hydrogen sulfide can be synthesized in the kidneys through the involvement of three enzymes: cystathionine γ-lyase, cystathionine β-synthase, and 3-mercaptopyruvate sulfurtransferase. These enzymes are localized in the glomeruli, tubulointerstitium, and tubular cells of the kidneys [42]. A low production of hydrogen sulfide has been associated with reduced expression of these metabolic enzymes in various kidney diseases [43]. Previous studies have demonstrated the multiple functions of hydrogen sulfide in kidney-related diseases, such as obstructive nephropathy, cisplatin-induced nephrotoxicity, diabetes-induced kidney injury, ischemic renal insult, and age-related alterations [17,44]. Hou et al. conducted an animal study to investigate the effect of exogenous H2S on age-related changes in structure and function [45]. The study showed that hydrogen sulfide alleviates the decline in renal function and collagen deposition in a dose-dependent manner. Specifically, the treated animals showed reduced oxidative stress and increased Nrf2 nuclear translocation. Lee et al. demonstrated that the administration of hydrogen sulfide restored AMPK activity and inhibited age-related signaling, such as the IR/IRS–Akt-mTORC1 axis. The experimental animals showed a reduced extracellular matrix deposition and improved renal function [46]. Because GYY4137 treatment was associated with an increase in SIRT1 and decrease in p21, we speculated that the AMPK and p53/p21 signal pathways were involved in ageing-related SGLT and GLUT alterations [33,46].

Anti-ageing therapy is very promising but presents great challenges. Learning to live healthily and maintain longevity is of utmost importance in medical society [47]. Lifestyle modifications such as exercise, weight control and diet interventions are the primary methods to achieve this, and are helpful and effective for reducing the risk of diabetes [8]. However, more and more pharmacological interventions have continued to evolve in recent decades, leading to lists of anti-inflammatory drugs and antioxidants. Most of them have demonstrated anti-ageing effects in vitro and in vivo but have limited clinical evidence. The discovery and development of senolytics and senomorphics, either natural or synthetic, are still ongoing and represent a novel approach to anti-ageing therapy [48]. Plants belonging to the *Alliaceae* botanical family, such as garlic, have long been used as an ingredient in cooking. Although the exact mechanism has not been completely explored, with its sustainable and slow-releasing hydrogen sulfide properties, garlic is considered to provide a therapeutic advantage for cardiovascular health [49].

Several study limitations need to be addressed. Firstly, one dose of hydrogen sulfide was examined in the present study. It remains unknown whether there is a differential dose response for various tissues or organs in aged animals. Further experiments would be helpful to clarify this issue. Secondly, an observational period of as long as 12 months was analyzed. The appropriate time required for hydrogen sulfide to intervene in ageing-related alterations in structure and function is not clear. Thirdly, the molecular mechanism(s) through which hydrogen sulfide modulates glucose transport was not explored. Future studies investigating this mechanism are needed.

## 4. Materials and Methods

### 4.1. Animals

Male Sprague Dawley rats were divided into different groups according to their age, including 3 month olds, 6 month olds, and 12 month olds. GYY4137, which releases hydrogen sulfide, was administered to animals for 8 weeks, after 16 and 40 weeks, in the 6-month and 12-month groups, respectively. The animals were grouped as follows: 3 months (*n* = 8); 6 months (*n* = 8); 12 months (*n* = 8); 6 months with normal chow and intraperitoneal injection of GYY4137 133μM/kg/day (Cayman Chemical, Ann Arbr, MI, USA, *n* = 8); 12 months with intraperitoneal injection of GYY4137 133 μM/kg/day (*n* = 8). We chose this dosage based on drug efficacy and safety [45,46,50].

### 4.2. Methods

Body weight and urine amount were measured every week until the end of the study. At the end of the study, animals were harvested, and their blood and urine samples were collected and measured. Blood levels of glucose, insulin, and creatinine were determined. Urinary levels of creatinine, total protein, glucose, and uric acid were also measured to calculate daily excretion using the SYNCHRON CX DELTA system (Beckman Coulter, Fullerton, CA, USA).

### 4.3. Kidney Tissue

After harvesting, kidney tissue was prepared for molecular studies, including PCR and protein blotting. The gene expression of ageing-related molecules, including SIRT1 and P21, was analyzed.

### 4.4. Pyruvate Tolerance Test

Pyruvate tolerance study was conducted one week before the end of study. After a 16 h fast, pyruvate (1.0 g/kg, Sigma-Aldrich, St. Louis, MO, USA) was intra-peritoneally injected into rats. Blood samples were collected at 30 min intervals to determine blood sugar levels after pyruvate injection for a total of 2 h.

### 4.5. Gene Expression Analysis

#### 4.5.1. RNA Isolation and cDNA Synthesis

Total RNA was extracted from kidney tissue using a Total RNA Mini kit, following the manufacturer’s instructions. Spectrophotometry was then used to detect the total RNA concentrations at 260 nm wavelengths. The RNA was stored at −80 °C for further use. A total of 1 μg RNA of from each sample was reverse-transcribed using the First Strand cDNA Synthesis Kit. Primers of each gene were listed as Table 3.

#### 4.5.2. Real-Time Polymerase Chain Reaction (PCR)

The alterations in gene expression were quantified by real-time polymerase chain reaction. The emission signal was assessed using the SYBR Green method (Applied Bioytems, Waltham, MA, USA). β-Actin was used as the internal reference for each evaluated gene. Specific primers for the investigated genes were then designed accordingly. To determine gene expression, genes studied in the present study were calculated as 2^(β-actin Ct-target gene Ct)^, where Ct represented the first cycle at which the output signal exceeded the threshold signal. The polymerase chain reaction for each gene was performed in duplicate to obtain a mean value. Changes in gene expression are presented as percentages of control animal values.

### 4.6. Hematoxylin and Eosin Staining

For histopathological examination, kidney tissues were fixed in 10% NBF, and 4 μm thick sections were stained with hematoxylin and eosin (H&E). The severity of glomerulosclerosis was assessed by calculating the percentage of glomeruli showing sclerosis and the extent of glomerulosclerosis within glomeruli using a grading system from 0 to 4. In brief, the glomeruli were graded from 0 to 4 as follows: grade 0, normal; grade 1, <25% involvement of the glomerular tuft; grade 2, 25–50% involvement of the glomerular tuft; grade 3, 50–75%; grade 4, with sclerosis occupying >75% of the glomerular tuft.

### 4.7. Protein Analysis (Immunoblotting)

Renal tissue samples were treated with a protein lysis buffer solution that contained 20 mM Tris-HCL (pH 7.4), 0.1% sodium dodecyl sulfate (SDS), 5 mM EDTA, 1% Triton X-100, and a protease inhibitor cocktail tablet (Thermo Fisher Scientific Waltham, MA, USA). After the concentration was determined, the protein samples were then run on 9% SDS-polyacrylamide gel electrophoresis (PAGE) for transfer to PVDF membranes. β-Actin was used as an internal control in this study. After complete blocking with 5% skim milk, specific antibodies for molecules, such as SGLT1/2, GLUT1/2, and klotho, were applied (SGLT1/2: 1:1000, ABclonal, MA, USA; SGLT2: 1:1000, Santa Cruz Biotechnology, CA, USA; GLUT1: 1:25,000, Abcam, Cambridge, UK; GLUT2: 1:1000, Proteintech, IL, USA; klotho: 1:1000, Proteintech, IL, USA). Finally, the membrane was incubated with the appropriate antibody in conjunction with an anti-rabbit IgG, HRP-linked antibody (1:10,000). All blots were taken by UVP BioSpectrum 810 Imaging System. The investigated molecules were quantified using densitometric analyses (Image J bundled with 64-bit Java 8, National Institute of Health, Bethesda, MD, USA). Changes in protein abundance are presented as percentages (%) of control animal values.

### 4.8. Statistical Analysis

Data analysis was performed using SPSS version 17 (IBM, Armonk, NY, USA). Biochemical data are expressed as mean ± standard deviation (SD), and the results of gene expression analysis and immunohistochemistry are displayed as mean ± SEM. Comparisons among groups were performed using ANOVA. A *p* < 0.05 was considered statistically significant.

## 5. Conclusions

In conclusion, our results indicated ageing-related alterations in renal epithelial glucose transporters. An increased HOMA-IR index and disturbed glucose homeostasis were also observed during the ageing process. The administration of exogenous GYY4137 lowered the HOMA-IR index. The increased expression of glucose transporters such as SGLT1/2 and GLUT1/2 was reversed by GYY4137.

## Figures and Tables

**Figure 1 ijms-24-16455-f001:**
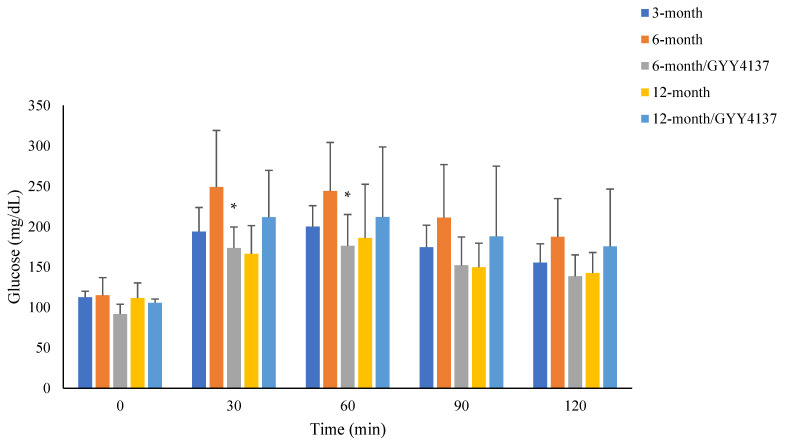
Blood glucose levels of different group animals at different time points in. pyruvate tolerance test (N = 6 in each group).

**Table 1 ijms-24-16455-t001:** Laboratory data of the animals in different groups.

	3-Month Group (N = 10)	6-Month Group (N = 8)	6-Month/ GYY4137 Group (N = 8)	12-Month Group (N = 8)	12-Month/ GYY4137 Group (N = 8)
Body weight (g)	592.3 ± 49.8	632.5 ± 65.5	595.5 ± 39.3	683.4 ± 70.4 *	693.8 ± 60.2 *
24 h urine, (mL)	20.3 ± 3.1	17.4 ± 3.9	25.3 ± 8.7	20.7 ± 7.0	21.0 ± 4.4
Blood glucose, (mg/dL)	104.0 ± 9.7	111.8 ± 5.3 *	108.3 ± 11.9 ^#^	118.7 ± 3.1 *	103.0 ± 10.7 ^##^
Blood insulin, (μU/mL)	2.5 ± 1.0	3.2 ± 1.9	3.3 ± 1.5	4.4 ± 1.1 *	3.5 ± 1.0 *
HOMA-IR	0.7 ± 0.2	0.6 ± 0.0 *	0.7 ± 0.2 *	1.2 ± 0.3 *	0.9 ± 0.2 *^##^
Blood creatinine, (mg/dL)	0.40 ± 0.04	0.47 ± 0.05 *	0.44 ± 0.04 ^#^	0.49 ± 0.06 *	0.45 ± 0.03 ^##^
Blood uric acid, (mg/dL)	1.0 ± 0.5	0.9 ± 0.8	0.8 ± 0.6	1.0 ± 0.4	0.7 ± 0.4
Creatinine clearance, (mg/min)	3.6 ± 0.6	3.2 ± 0.3	3.4 ± 0.5	3.4 ± 0.6	3.7 ± 0.3
Urine glucose excretion, (mg/24 h)	53.2 ± 23.8	54.1 ± 29.4	50.8 ± 23.5	58.5 ± 23.5	52.3 ± 16.8
Urine uric acid excretion, (mg/24 h)	16.2 ± 3.8	15.2 ± 3.9	18.0 ± 2.3	23.9 ± 6.7 *	24.8 ± 3.7 *
Urine protein excretion, (mg/24 h)	58.7 ± 25.6	71.2 ± 16.0 *	52.0 ± 11.9	87.1 ± 24.4 *	52.5 ± 18.2 ^##^

* *p* < 0.05 vs. 3-month group, ^#^ *p* < 0.05, 6-month group vs. G6-month group. ^##^ *p* < 0.05, 12-month group vs. G12-month group.

**Table 2 ijms-24-16455-t002:** Glomerulosclerosis index in study animals.

**Age**	3-M	6-M	G6M	12-M	G12M
**Grade**	0	0	0	1	0-1

**Table 3 ijms-24-16455-t003:** The primer sequences of studied genes.

Gene	Forward (5′-3′)	Reverse (5′-3′)
β-actin	AGTACCCCATTGAACACGGC	TTTTCACGGTTGGCCTTAGG
P21 (*Cdkn1a*)	CACACAGGAGCAAAGTATG	TCAAAGTTCCACCGTTCT
Sirtuin 1 (*SIRT1*)	CCTCTGCCTCATCTACATT	TACTCGCCACCTAACCTA

## Data Availability

The data presented in this study is available on request from the corresponding author.

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
