# Peer review of "Ageing-Related Alterations in Renal Epithelial Glucose Transport"

_ijms, 2023, doi:10.3390/ijms242216455_

Round 1
Reviewer 1 Report
Comments and Suggestions for Authors
This is a well-designed study that provides new information in the timely and important area of renal epithelial glucose transport in aging. It's findings support the current use of inhibitors of glucose transport in specific disorders of the kidney and glucose metabolism.
An interesting observation that there was no significant change in daily urinary glucose excretion in the 6- and 12-month group comparing with the 3-month group. They state that "as renal threshold for glucose increases with age, glucosuria is not evident in the elderly and thus polyuria, polydipsia may not become obvious in the elderly. "
This need further explanation in the discussion.
Author Response
Response: (1) Thank you for your comment. Our results indicated there was no significant increase in urinary glucose excretion during the ageing process of study animals despite the fact that blood glucose concentration was increased. As we mentioned that the development of hyperglycemia or diabetes in the elderly is caused by a variety of metabolic factors, the increased expression of SGLT1/2 might enhance tubular glucose reabsorption, thus increased systemic glucose loading. Nevertheless, the overall effect on urinary glucose excretion is probably neutral, not necessarily increased or decreased. (2) On the other hand, Kalra et al. (reference 18) pointed out renal threshold for glucose is increased with age and thirst mechanisms are impaired, the symptoms of polyuria and polydipsia may not present early in elderly persons with diabetes. In this paper, the mechanism responsible for change in renal glucose threshold was not explained. Currently, it is unclear whether increased tubular reabsorption as noted in our study contributes to the increased renal threshold, but we agree that changes in symptomology deserve clinicians’ attention. We have added these supplements in our revised manuscript.
Reviewer 2 Report
Comments and Suggestions for Authors
This study entitled "Ageing-related Alterations in Renal Epithelial Glucose Transport" by Lee CT et al. was designed to investigate the effect of hydrogen sulfide donor (GYY4137) on the renal epithelial glucose transporters in male Sprague Dawley rats in the context of glucose metabolism ageing-related changes in glucose metabolism. The study reported that blood glucose levels and HOMA-IR indices were elevated in the 12-month group of rats, while GYY4137 reduced these indices. This was associated with the expression of SGLT1, SGLT2, GLUT1 and GLUT2. The authors conclude that GYY4137 is effective in reversing the glucose transporter and homeostatic alternations associated with aging.
Specific comments:
(1) The protective effects of hydrogen sulfide on aging kidneys and the ability of hydrogen sulfide to ameliorate changes associated with renal aging are well established (PMID: 27882191 and 29717417). What was the authors' rationale for choosing SD rats? Furthermore, are the same changes observed in female SD rats? The authors compared the experimental group to the 3-month group. Since the 12-month group was 6 months older than the 6-month group, the 12-month group also needs to be compared to the 6-month group.
(2) This study requires the detection of H2S levels in plasma, urine and kidney as well as enzyme expression of cystathionine γ-lyase (CSE), cystathionine β-synthase (CBS) and 3-mercapto pyruvate sulfotransferase (3MST).
(3) Why use 133uM/kg/day as the working concentration of GYY4137 and ensure that GYY4137 is effective in the kidney?
(4) Can the authors explain why there is a difference in the two phenotypes at 2 months of GY4137 treatment, given that the 6-month/GYY4137-treated group lost weight compared to the 6-month group and the 12-month/GYY4137-treated group gained weight compared to the 12-month group? Moreover, please provide longitudinal body weight measurement as weight is monitored weekly.
(5) The authors should explain why urinary glucose excretion was higher in the 6- and 12-month groups than in the 6- and 12-month/GYY4137 groups, respectively, in the presence of markedly increased expression of SGLT2 and SGLT1, and, conversely, lower urinary glucose excretion of down-regulated SGLT1 and SGLT2 in the GYY4137-treated 6- and 12-month groups.
(6) Figure 1, the pyruvate tolerance test should incorporate area under the curve.
(7) Figure 2, Glomerulosclerosis should be quantified by Masson trichrome staining. In addition, the number of podocytes should be counted.
(8) Figure 3, the statistics labels should be aligned with the groups. Detection of protein expression of SIRT1 and P21 by immunoblotting or immunohistochemistry (IHC) should include.
(9) Figure 4, protein expression level of SGLT1, SGLT2, GLUT1 and GLUT2 co-stained with proximal tubular marker and distal tubular marker by immunofluorescence (IF) should provide.
(10) The authors could reduce the volume in the introduction and discussion on the pancreas, liver, skeletal muscle and adipose tissue, because the authors did not further elucidate the effects of GYY4137 on these tissues and provide evidence to dispel the claim that GYY4137 has no effect on these tissues. Due to the limitations of this study, the authors did not explore the molecular mechanisms of how GYY4137 regulates those glucose transporters, and therefore the authors should review and present the possible mechanisms that may activate or inhibit the expression of the glucose transporters, such as transcription factors, in the discussion section.
(11) Please check the citation for reference 1 in line 28, and reference 4 in the line 4 which are not appropriate.
(12) Indent of the paragraph is random, line 73, line 214, line 219, line 234, line 374 and line 375.
Comments on the Quality of English LanguageMinor editing of English language required
Author Response
Thank you for your suggestions and comments. We have revised our manuscript and uploaded the revised manuscript.

Round 2
Reviewer 2 Report
Comments and Suggestions for Authors
Thanks to the author for his reply, however, some concerns were not clearly explained. Revised comments are highlighted.
Specific comments:
(1) The protective effects of hydrogen sulfide on aging kidneys and the ability of hydrogen sulfide to ameliorate changes associated with renal aging are well established (PMID: 27882191 and 29717417). What was the authors' rationale for choosing SD rats? Furthermore, are the same changes observed in female SD rats? The authors compared the experimental group to the 3-month group. Since the 12-month group was 6 months older than the 6-month group, the 12-month group also needs to be compared to the 6-month group.
Response: (1) Previous animal experiments on ageing-related issues have been tested in rats, mice or even other animals. In the present study, we only examined changes of renal epithelial glucose transporters in rats, this does not necessarily indicate other species would not have these changes. SD rats were selected as our Lab is familiar with these animals. We only tested on male rats and it remains unclear whether our findings in the present study can be extended in female rats until been proved. (2) Yes, we agree that comparison among different age groups is indicated and can provide more information. In our analysis, ANOVA with post hoc Scheffe test was used to find out any difference among groups, including between any two groups. The results were shown in table 1, and were remarked significant if p<0.05.
Comment: (A) The authors do not state why H2S levels in the kidneys of Sprague Dawley rats decrease with age. The basis for this argument must be substantiated. Regarding specific point (1), and the following points (2) and (3), the authors should verify if H2S levels are indeed age-related, and also, in the current study, does administering GYY4137 increase H2S levels? (B) The authors noted in Table 1 that they also compared the 12-month group with the 6-month group and remarked it if P<0.05. However, there was a significant difference in the HOMA-IR results in the 12-month group (1.2±0.3) compared with the 3-month group (0.7±0.2), whereas there was no difference in the 12-month group (1.2±0.3) compared with the 6-month group (0.6±0.0). Please re-verify all statistical analyses.
(2) This study requires the detection of H2S levels in plasma, urine and kidney as well as enzyme expression of cystathionine γ-lyase (CSE), cystathionine β-synthase (CBS) and 3-mercapto pyruvate sulfotransferase (3MST).
Response: (1) We did not measure H2S levels in blood and urine samples in the present study. Previous studies have demonstrated that ageing was associated with reduced H2S level in both clinical investigation and experimental animal models. Exogenous administration of H2S such as GYY4137 can reverse the decrease. We plan to develop the method and detect these changes in future relevant research to obtain comparable results. (2) As we mentioned in discussion, these three key enzymes are expressed in renal tissue, including glomeruli, tubular cells and interstitum. A variety of kidney diseases, including ageing, diabetes are associated with decreased expression of these enzymes, indicating a reduced H2S production. In our experiment, the decrease in sirtuin 1 combined with increase in p21 in renal tissue represent an ongoing ageing process. We used this finding to indicate kidney ageing. Treatment effect of GYY4137 was documented RT-PCR results of these two molecules.
Comment: In the current study, does the expression of these enzymes (CSE, CBS, and 3MST) change in non-disease models and age-related in SD rats? GYY4137 is not a sirtuin1 activator nor a p21 inhibitor, thus the change of sirtuin1 and p21 gene expression are indirect via GYY4137.
(3) Why use 133uM/kg/day as the working concentration of GYY4137 and ensure that GYY4137 is effective in the kidney?
Response: Thank you for your comment. Previous studies had been using different doses of exogenous hydrogen sulfide to reverse ageing processes. The range of administration dose is rather wide (10-200 μmol/kg/day, Hou et al: Oxid Med Cell Longev 2016,7570489; Lin et al: J Urolog 2016, 16, 1778-1787; Lee at al: GeroScience 2018, 40: 163-176.). We took this range as our reference to prepare treatment dose in the present study. This dose (133μmol/kg/day) did not induce toxic effects on observed organs (kidney, liver). Nevertheless, it remains unclear whether different organs respond to similar dose or not. More advanced study is indicated to answer this question.
Comment: Please incorporate this judgment and reference into the methodology section.
(4) Can the authors explain why there is a difference in the two phenotypes at 2 months of GY4137 treatment, given that the 6-month/GYY4137-treated group lost weight compared to the 6-month group and the 12-month/GYY4137-treated group gained weight compared to the 12-month group? Moreover, please provide longitudinal body weight measurement as weight is monitored weekly.
Response: In terms of comparison among 5 animal groups, one-way ANOVA with post hoc Scheffe test was used to examine if any significant difference. As shown in table 1, significant difference was observed in 12-month group animals comparing with 3-month group irrespective of GYY4137 treatment. There was no significant difference between 6-month group animals with/without GYY4137 treatment. Similarly, no difference was observed between 12-month groups between with/without GYY4137 treatment.
Comment: Why there is a difference in the two phenotypes at 2 months of GY4137 treatment, given that the 6-month/GYY4137-treated group lost weight compared to the 6-month group and the 12-month/GYY4137-treated group gained weight compared to the 12-month group? Because body weight was measured weekly until the end of the study. Please provide the body weight progression in this study.
(5) The authors should explain why urinary glucose excretion was higher in the 6- and 12-month groups than in the 6- and 12-month/GYY4137 groups, respectively, in the presence of markedly increased expression of SGLT2 and SGLT1, and, conversely, lower urinary glucose excretion of down-regulated SGLT1 and SGLT2 in the GYY4137-treated 6- and 12-month groups.
Response: (1) The increased expression of SGLT1/2 in our animal model may enhance tubular glucose reabsorption and thus leads to increased systemic glucose load and elevate blood glucose concentration. On the other hand, in the face of more filtered glucose via glomeruli enters renal tubules, urinary glucose excretion might be increased. Moreover, other systemic factors can also affect renal glucose handling. Therefore, it is difficult to make a clear conclusion regarding urinary glucose excretion. (2) As we mentioned previously, table 1 displays result of statistic analysis by using ANOVA. There was no significant difference in daily urinary glucose excretion among the 5 animal groups.
Comment: Whether higher systemic glucose load also increases expression of SGLT1/2. With age, kidney size increases and the S1/S2 renal proximal tubules become longer. Did the authors note a difference in kidney size with age and GYY4137 treatment?
(6) Figure 1, the pyruvate tolerance test should incorporate area under the curve.
Response: Thank you for your comment. The pyruvate tolerance test was performed to examine hepatic glucose production during ageing process in this experiment. Moreover, we examineed whether GYY4137 treatment has any effect on the liver. Comparison among different age groups and different time points were also analyzed. Therefore, we did not summarize all data into AUC.
Comment: The current Figure 1 is not easy to follow, please provide the AUC or show the different time points in a bar graph.
(7) Figure 2, Glomerulosclerosis should be quantified by Masson trichrome staining. In addition, the number of podocytes should be counted.
Response: (1) Both periodic acid-schiff (PAS) staining and hematoxylin and eosin (H&E) staining are commonly used for renal histology study. In renal pathology PAS stain is particularly useful to highlight basement membranes and matrix. The hematoxylin stains cell nuclei purple, and eosin stains the extracellular matrix as well as the cytoplasm pink, giving an overall impression tissue morphology. Glomerulosclerosis index is an important maker for kidney disease as well as renal ageing. Since our animal model represents early, not advanced stage of ageing, despite biochemical and functional changes were observed, structural change is not apparent. In our pathological assessment, glomeruli were mostly intact, and glomerulosclerosis index was not high under H&E staining. (2) To count podocyte number during ageing process is a very interesting issue. One recent review article from Shankland et al. (JASN 2021, 32: 2697-2613) clearly indicated podocyte ageing and depletion plays an important role in kidney healthy ageing and pre-existing renal diseases. This extensive review also highlights podocyte-specific ageing mechanism and signal pathways. Podocyte number in glomeruli was not measured in this experiment. We hope our Lab can develop the method and advance our future study in kidney ageing.
Comment: Okay.
(8) Figure 3, the statistics labels should be aligned with the groups. Detection of protein expression of SIRT1 and P21 by immunoblotting or immunohistochemistry (IHC) should include.
Response: (1) We have corrected this error, thank you. (2) Only RT-PCR was performed to investigate the gene expression changes of SIRT1 and P21 in our study.
Comment: The function of SIRT1 and P21 lies at the protein level. Can mRNA of SIRT1 and P21 represent protein expression?
(9) Figure 4, protein expression level of SGLT1, SGLT2, GLUT1 and GLUT2 co-stained with proximal tubular marker and distal tubular marker by immunofluorescence (IF) should provide.
Response: Immunofluorescence study can offer protein abundance of these molecules in specific segment(s) of renal tubules. Because expression of above glucose transporters (SGLT1/2, GLUT1/2) are mostly confined to proximal tubule, immunoblot study is an alterative method to determine the changes of protein abundance in renal tissue. We understand that GLUT1 is found at various levels in virtually all nephron segment, in which it enables glucose influx into cells according to nutritional requirement. It is with regret that right now we do not have immunofluorescence study to provide in this study.
Comment: Okay.
(10) The authors could reduce the volume in the introduction and discussion on the pancreas, liver, skeletal muscle and adipose tissue, because the authors did not further elucidate the effects of GYY4137 on these tissues and provide evidence to dispel the claim that GYY4137 has no effect on these tissues. Due to the limitations of this study, the authors did not explore the molecular mechanisms of how GYY4137 regulates those glucose transporters, and therefore the authors should review and present the possible mechanisms that may activate or inhibit the expression of the glucose transporters, such as transcription factors, in the discussion section.
Response: (1) Because ageing per se affect glucose homeostasis. We made a brief review in organs that play fundamental role in glucose metabolism such as pancreas, liver, muscle and adipose tissue in addition to kidney. In our revised manuscript, we have deleted these descriptions and just kept important part. The liver part was preserved because pyruvate test was performed to examine hepatic glucose production during ageing process. (2) Yes, the molecular regulatory mechanism was not investigated in the present study. We have listed this as study limitation in our manuscript. With regard to the probable mechanism(s), we have added the following supplements: because GYY4137 treatment was associated with increase in SIRT1 and decrease in p21, we speculated that at least AMPK and p53/p21 signal pathways were involved in ageing-related SGLT and GLUT alterations.
Comment: To support this hypothesis, please provide relevant references or evidence.
(11) Please check the citation for reference 1 in line 28, and reference 4 in the line 4 which are not appropriate.
Response: We have replaced reference 1 by other paper; one additional paper was cited to illustrate the ageing-associated altered glucose metabolism (reference 4 and 5).
Comment: Okay.
(12) Indent of the paragraph is random, line 73, line 214, line 219, line 234, line 374 and line 375.
Response: We have corrected these errors, thank you.
Comment: Okay.
Comments on the Quality of English LanguageMinor editing of English language required
Author Response
Dear Reviewer:
Thanks for your comments on our submission. According to these comments and suggestions, we have revised our manuscript substantially and it has been uploaded as a new file (revision version 2). Please notify us if further work is needed for acceptance by your journal.
Thank you!

Round 3
Reviewer 2 Report
Comments and Suggestions for Authors
The authors have answered the reviewer's comments and improved the manuscript. I have no further comments.
Comments on the Quality of English LanguageMinor editing of English language required